# CAJun: Continuous Adaptive Jumping using a Learned Centroidal Controller

**Yuxiang Yang**[*], **Guanya Shi**[†], **Xiangyun Meng**[*], **Wenhao Yu**[‡], **Tingnan Zhang**[‡]
**Jie Tan**[‡], **Byron Boots**[*]
[*]University of Washington   [†]Carnegie Mellon University   [‡]Google Deepmind

**Abstract:** We present CAJun, a novel hierarchical learning and control framework that enables legged robots to jump continuously with adaptive jumping distances. CAJun consists of a high-level centroidal policy and a low-level leg controller. In particular, we use reinforcement learning (RL) to train the centroidal policy, which specifies the gait timing, base velocity, and swing foot position for the leg controller. The leg controller optimizes motor commands for the swing and stance legs according to the gait timing to track the swing foot target and base velocity commands. Additionally, we reformulate the stance leg optimizer in the leg controller to speed up policy training by an order of magnitude. Our system combines the versatility of learning with the robustness of optimal control. We show that after 20 minutes of training on a single GPU, CAJun can achieve continuous, long jumps with adaptive distances on a Go1 robot with small sim-to-real gaps. Moreover, the robot can jump across gaps with a maximum width of 70cm, which is over $40\%$ wider than existing methods.[1]

**Keywords:** Jumping, Legged Locomotion, Reinforcement Learning

## 1 Introduction

Legged robots possess a unique capability to navigate some of the earth's most challenging terrains. By strategically adjusting their foot placement and base pose, legged robots can negotiate steep slopes [1, 2, 3], traverse uneven surfaces [4, 5], and crawl through tight spaces [6]. However, for terrains with scarce contact choices, such as gaps or stepping stones, the capability of legged robots remains somewhat limited. This limitation primarily stems from the fact that most legged robots rely heavily on walking gaits with continuous foot contacts. As such, options for foot placement are confined to within one body length from the robot's current location. Jumping offers a compelling solution to this problem. By enabling "air phases", a jumping robot can traverse through long distances without terrain contacts. Such a capability could markedly enhance a legged robot's versatility when dealing with challenging terrains. In addition, a robot capable of *continuous*, *adaptive* and *long-distance* jumps could further boost its speed and efficiency during terrain traversal.

Compared with standard walking, jumping is a significantly more challenging control task for both optimization-based [7, 8, 9, 10, 11, 6, 12, 13, 14] and learning-based controllers [15, 5, 16, 17, 18]. Optimization-based controllers, despite proving robust in challenging terrains, face computational limitations that prevent them from planning for long jumping trajectories in real time. Typically, these controllers circumvent this issue by first solving an intricate trajectory optimization problem offline, then utilizing simplified model predictive control (MPC) to track this predetermined *fixed* trajectory online. Consequently, existing works tend to be restricted to non-adaptive, single jumps [6, 12, 13, 14]. On the other hand, RL controllers have the potential to learn more adaptive and versatile locomotion skills, but they require substantial effort in reward design and sim-to-real transfer [19, 15, 20, 4], particularly for dynamic and underactuated tasks such as jumping. Therefore, achieving continuous jumping over long distances can be a significant challenge for existing methods.

---

[1]Video and code at this page. Author Emails: {yuxiangy,xiangyun,bboots}@cs.washington.edu, guanyas@andrew.cmu.edu, {magicmelon,tingnan,jietan}@google.com

7th Conference on Robot Learning (CoRL 2023), Atlanta, USA.

In this paper, we present CAJun (Continuous Adaptive Jumping with a Learned Centroidal Policy), which achieves continuous long-distance jumpings with adaptive distances on the real robot. Our framework seamlessly combines optimization-based control and RL in a hierarchical manner. Specifically, a high-level RL-based *centroidal policy* specifies the desired gait, target base veloctiy, and swing foot positions to the *leg controller*, and a low-level *leg controller* solves the optimal motor commands given the centroidal policy's action. Our framework effectively integrates the benefits of both control and learning. First, the RL-based *centroidal policy* is able to learn versatile, adaptive jumping behaviors without heavy computational burden. Second, the low-level quadratic-programming-based (QP) *leg controller* optimizes torque commands at high frequency (500Hz), which ensures reactive feedback to environmental perturbations and significantly reduces the sim-to-real gap. Finally, to resolve the common training speed bottleneck in hierarchical methods [21, 22, 23], we reformulated the QP problem in the *leg controller* to a least-squares problem with clipping so that the entire stack is 10 times faster and can be executed in massive parallel [16].

Within 20 mins of training in simulation, we deploy CAJun directly to a Unitree Go1 robot [24]. Without any fine-tuning, CAJun achieves continuous, long-distance jumping, and adapts its jumping distance based on user command. Moreover, using the alternating contact pattern in a bounding gait, the robot is capable of crossing a gap of 70cm, which is at least 40% larger than existing methods (Fig. 4 and Table 1). To the best of our knowledge, CAJun is the first framework that achieves continuous, adaptive jumping with such gap-crossing capability on a commercially available quadrupedal robot. We further conduct ablation studies to validate essential design choices. In summary, our contribution with CAJun are the following:

- We present CAJun, a hierarchical learning and control framework for continuous, adaptive, long-distance jumpings on legged robots.

- We demonstrate that jumping policies trained with CAJun can be directly transferred to the real world with a gap-crossing capability of 70cm.

- We show that CAJun can be trained efficiently in less than 20 minutes using a single GPU.

## 2   Related Works

**Optimization-based Control for Jumping**   Using optimization-based controllers, researchers have achieved a large variety of jumping behaviors, from continuous pronking and bounding [7, 8, 9, 10, 11] to large single-step jumps [6, 12, 13, 14]. By optimizing for control inputs at a high frequency, these controllers can execute robust motions even under severe perturbations [8, 9]. However, due to the high computation cost, they cannot plan ahead for a long horizon during online execution. Therefore, they primarily focus on high-frequency jumps with a short CoM displacement per jump [9, 10, 11]. One way to overcome this computation limit is to pre-compute a reference trajectory offline using trajectory optimization (TO) [6, 12, 13, 14], which can greatly extend the height [12] and distance [13] of each jump. However, it can be challenging to generalize beyond the reference trajectories towards continuous, adaptive jumping [25, 26, 27]. Notably, using a multi-level planner, Park et al. [26] achieved continuous bounding with fixed gait and adaptive height to jump over hurdles. Compared to these approaches, our framework adopts a more general formulation, where the policy can adjust the gait timing, base pose, and swing foot position simultaneously.

**Learning-based Control for Jumping**   In recent years, learning-based controllers have significantly improved the capability of legged robots, from rapid running [28] to traversing over challenging terrains [5]. While standard walking gaits can be learned from scratch using reinforcement learning (RL), more dynamic behaviors such as jumping usually require additional setup in the learning process, such as motion imitation [19, 18, 17], curriculum learning [16] and multi-stage training [3, 29]. Another challenge for learning-based controllers is sim-to-real, especially for dynamic underactuated behaviors like jumping [30]. To overcome the sim-to-real gap, researchers have developed a suite of tools such as domain randomization [15], system identification [31] and motor adaptation [20]. Recently, Smith et al. [19] used motion imitation and transfer learning to jump

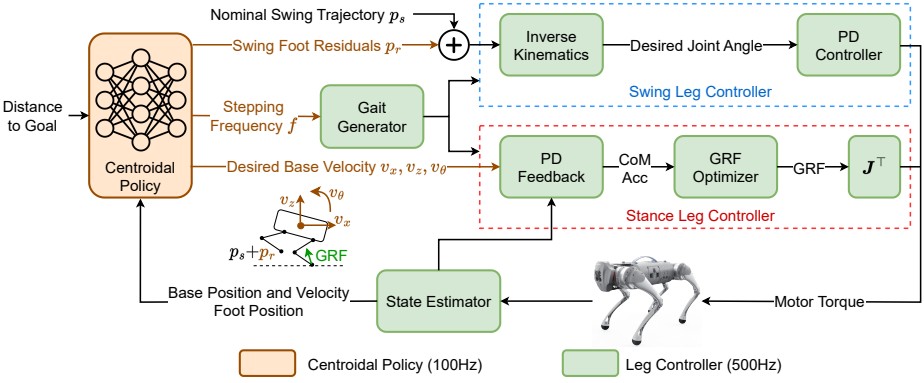

Figure 1: Overview of the hierarchical framework of CAJun.

over a gap of 20cm (0.4 body length) on a Unitree A1 robot, and Caluwaerts et al. [3] used multi-stage training with policy synthesis to jump over a gap of 50cm (1 body length) on a custom-built quadrupedal robot. Compared to these works, CAJun's hierarchical setup can jump over *wider* gaps (70cm / 1.4 body length) *continuously*, and can adapt its landing position based on user command.

**Hierarchical RL for Legged Robots**    Recently, there has been increasing interest in combining RL with optimization-based control for legged robots [22, 32, 21, 23, 33, 34]. These frameworks typically follow a hierarchical structure, where a high-level RL-trained policy outputs intermediate commands to a low-level leg controller. The RL policy can give several forms of instructions to the low-level controller, such as gait timing [22, 32], CoM trajectory [21, 34, 30, 33] and foot landing positions [23, 35, 36, 37, 38]. Our approach uses a similar hierarchical setup but adopts a general action space design where the policy specifies the gait, CoM velocity and swing foot locations *simultaneously*. One bottleneck of the hierarchical approaches is the slow training time because every environment step involves solving the optimization problem in the low-level controller. We overcome this bottleneck by relaxing the constraints in foot force optimization [39, 40, 41], so that foot force can be solved efficiently in closed form. Compared to existing frameworks which can take hours or even days to train, CAJun can be trained in 20 minutes using GPU-accelerated simulation [16].

## 3    Overview of CAJun

In order to learn continuous, long-distance, and adaptive jumping behaviors, we design CAJun as a hierarchical framework consisting of a high-level centroidal policy and a low-level leg controller (Fig. 1). To specify a jump, The *centroidal policy* outputs three key actions to the low-level controller, namely, the stepping frequency, the swing foot residual, and the desired base velocity. The modules in the *leg controller* then convert these actions into motor commands. Similar to previous works [8, 9, 32, 22, 21], the *leg controller* adopts separate control strategy for swing and stance legs, where the desired contact state of each leg is determined by the *gait generator*. We design the gait generator to follow a pre-determined contact sequence with timings adjustable by the high-level centroidal policy. For swing legs, we first find its desired position based on a heuristically-determined reference trajectory and learned residuals, and converts that to joint position commands using inverse kinematics. For stance legs, we first determine the desired base acceleration from the policy commands, and then solves an optimization problem to find the corresponding Ground Reaction Forces (GRFs) to reach this acceleration. We run the low-level controller at 500Hz for fast, reactive torque control, and the high-level controller at 100Hz to ensure stable policy training.

## 4    Low-level Leg Controller

Similar to prior works [22, 21, 8], the low-level controller of CAJun adopts separate control strategies for swing and stance legs, and uses a gait generator to track the desired contact state of each leg. Additionally, we carefully design the interface between the centroidal policy and components in the leg controller to maintain control robustness and policy expressiveness. Moreover, we relaxed the GRF optimization problem in stance leg controller to significantly speed up training.

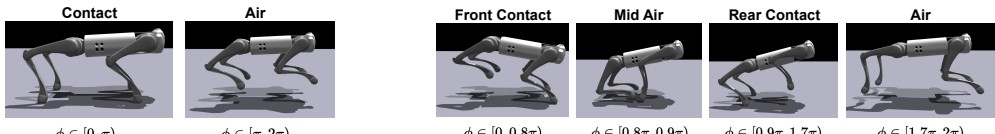

| Contact | Air | Front Contact | Mid Air | Rear Contact | Air |
|---|---|---|---|---|---|
| $\phi \in [0, \pi)$ | $\phi \in [\pi, 2\pi)$ | $\phi \in [0, 0.8\pi)$ | $\phi \in [0.8\pi, 0.9\pi)$ | $\phi \in [0.9\pi, 1.7\pi)$ | $\phi \in [1.7\pi, 2\pi)$ |

Figure 2: The contact sequence and default timing of the pronking (**left**) and bounding (**right**) gait.

## 4.1 Phase-based Gait Generator

The gait generator determines the desired contact state of each leg (swing or stance) based on a pre-defined contact sequence and the timing information from the centroidal policy. To capture the cyclic nature of locomotion, we adopt a phase-based gait representation, similar to prior works [22, 42]. The gait is modulated by a phase variable $\phi$, which increases monotonically from 0 to $2\pi$ in each locomotion cycle, and wraps back to 0 to start the next cycle. The propagation of $\phi$ is controlled by the *stepping frequency* $f$, which is commanded by the centroidal policy:

$$\phi_{t+1} = \phi_t + 2\pi f \Delta t \tag{1}$$

where $\Delta t$ is the control timestep. The mapping from $\phi$ to the desired contact state is pre-defined. We adopt two types of jumping gaits in this work, namely, *bounding* and *pronking*, where bounding alternates between the front and rear leg contacts, and pronking lands and lifts all legs at the same time (Fig. 2). Note that while the *sequence* of contacts is fixed, the centroidal policy can flexibly adjust the *timing* of contacts to based on the state of the robot.

## 4.2 Stance Leg Control

The stance leg controller computes the desired joint torque given the velocity command from the centroidal policy. Since jumping is mostly restricted to the sagittal plane, the policy specifies the velocity in the forward and upward axis $(v_x, v_z)$, as well as the rotational velocity $v_\theta$, and the velocity for the 3 remaining DoF is set to 0. We compute the desired torque following a 3-step procedure. First, we compute the desired CoM acceleration $\ddot{\boldsymbol{q}}^{\text{ref}} \in \mathbb{R}^6$ using a PD controller (Appendix. A.2). Next, we optimize for the GRF $\boldsymbol{f} = [\boldsymbol{f}_1, \boldsymbol{f}_2, \boldsymbol{f}_3, \boldsymbol{f}_4] \in \mathbb{R}^{12}$ to track this desired acceleration, where $\boldsymbol{f}_i$ is the foot force vector of leg $i$. Lastly, we compute the motor torque command using $\boldsymbol{\tau} = \boldsymbol{J}^\top \boldsymbol{f}$, where $\boldsymbol{J}$ is the foot Jacobian. When training a hierarchical controller with a low-level optimization-based controller, a major computation bottleneck lies in the GRF optimization [21, 22, 23]. As such, we re-design this optimization procedure to significantly speed up the training process.

**QP-based GRF Optimization**  To optimize for GRF, prior works typically solve the following quadratic program (QP):

$$\min_{\boldsymbol{f}} \|\ddot{\boldsymbol{q}} - \ddot{\boldsymbol{q}}^{\text{ref}}\|_{\mathbf{U}} + \|\boldsymbol{f}\|_{\mathbf{V}} \tag{2}$$

$$\text{subject to: } \ddot{\boldsymbol{q}} = \mathbf{A}\boldsymbol{f} + \boldsymbol{g} \tag{3}$$

$$f_{i,z} = 0 \qquad\qquad\qquad \text{if } i \text{ is a swing leg} \tag{4}$$

$$f_{\min} \le f_{i,z} \le f_{\max} \qquad\qquad \text{if } i \text{ is a stance leg} \tag{5}$$

$$-\mu f_{i,z} \le f_{i,x} \le \mu f_{i,z}, \quad -\mu f_{i,z} \le f_{i,y} \le \mu f_{i,z} \qquad i = 1, \dots, 4 \tag{6}$$

Eq. (3) represents the centroidal dynamics model [8], where $\mathbf{A}$ is the generalized time-variant inverse inertia matrix, and $\boldsymbol{g}$ is the gravity vector (see Appendix. A.2 for details). Eq. (5), (4) specifies the contact schedule, as computed by the gait generator. Eq. (6) specifies the approximated friction cone constraints, where $\mu$ is the friction coefficient. $\mathbf{U}, \mathbf{V} \succ 0$ are positive definite weight matrices.

**Unconstrained GRF Optimization with Clipping**  QP-based GRF optimization would require an iterative procedure (e.g., active set method or interior point method), which can be computationally expensive and difficult to scale up in parallel in GPU. Instead of using the QP formulation, CAJun relaxes this optimization problem by solving the unconstrained GRF first and clipping the resulting GRF to be within the friction cone. Since Eq. (3) is linear in $\boldsymbol{f}$, if we ignore the constraints in Eq. (6) and (5) and eliminate the variables for non-contact legs, the optimal $\boldsymbol{f}$ can be solved *in closed-form*:

$$\widehat{\boldsymbol{f}} = (\mathbf{A}^\top \mathbf{U} \mathbf{A} + \mathbf{V})^{-1} \mathbf{A}^\top \mathbf{U} (\ddot{\boldsymbol{q}}^{\text{ref}} - \boldsymbol{g}) \tag{7}$$

Next, we project the solved ground reaction forces into the friction cone, where we first clip the normal force within actuator limits, and then clip the tangential forces based on the clipped normal force:

$$f_{i,z} = \text{clip}(\widehat{f}_{i,z}, f_{\min}, f_{\max}), \quad (f_{i,x}, f_{i,y}) = (\widehat{f}_{i,x}, \widehat{f}_{i,y}) \cdot \min(1, \mu f_{i,z}/\sqrt{\widehat{f}_{i,x}^2 + \widehat{f}_{i,y}^2}) \tag{8}$$

We design this projection to minimize the force disruption in the gravitational direction, so that the low-level controller can track height commands accurately.

Note that our unconstrained formulation not only reduces computational complexity, but also makes the solving procedure highly parallelizable. Therefore, when paired with GPU-accelerated simulator like Isaac Gym [16], CAJun can be trained efficiently in massive parallel, which significantly reduced the turn-around time. Additionally, while our unconstrained formulation may yield sub-optimal solutions when the least-squares solution (Eq. (7)) finds a GRF outside the friction cone, the high-level *centroidal policy* would observe this sub-optimality during training, and thereby compensating for it by adjusting the desired CoM velocity commands. In practice, we find the policies trained using the constrained and unconstrained optimization to perform similarly (see Sec. 6.6 for details).

### 4.3   Swing Leg Control

We use position control for swing legs, where the desired position is the sum of a heuristically constructed reference trajectory [8, 43] and a residual output from the centroidal policy. Similar to prior works [22, 21], we generate the reference trajectory by interpolating between key points in the swing phase (see Appendix. A.3 for details). On top of the heuristic trajectory ($\boldsymbol{p}_s$ in Fig. 1), the centroidal policy adjusts the swing foot trajectory for higher foot clearance and optimal foot placement by outputting a residual in foot position ($\boldsymbol{p}_r$ in Fig. 1). Once the foot position is determined, we convert it to desired motor angles using inverse kinematics and execute it using joint PD commands.

## 5   Learning a Centroidal Policy for Jumping

The RL problem is represented as a Markov Decision Process (MDP), which includes the state space $\mathcal{S}$, action space $\mathcal{A}$, transition probability $p(s_{t+1}|s_t, a_t)$, reward function $r : \mathcal{S} \times \mathcal{A} \mapsto \mathbb{R}$, and initial state distribution $p_0(s_0)$. We aim to learn a policy $\pi : \mathcal{S} \mapsto \mathcal{A}$ that maximizes the expected cumulative reward over an episode of length $T$, which is defined as $J(\pi) = \mathbb{E}_{s_0 \sim p_0(\cdot), s_{t+1} \sim p(\cdot|s_t, \pi(s_t))} \sum_{t=0}^{T} r(s_t, a_t)$.

**Environment Overview**   For maximum expressiveness, we design the environment such that the centroidal policy directly specifies the contact schedule, base velocity and swing foot position for the low-level controller. To focus on continuous jumps, we design each episode to contain exactly 10 jumping cycles, where termination is determined by the gait generator (Section. 4.1). Additionally, we normalize the reward so that total reward within each jumping cycle is agnostic to its duration. In order to learn distance-adaptive jumping, we sample different jumping distances uniformly in [0.3m, 1m] before each jump, and compute the desired landing position, which is included in the state space.

**State and Action Space**   We design the state space to include the robot's proprioceptive state, as well as related information about the current jump. The proprioceptive information includes the current position and velocity of the robot base, as well as the foot positions in the base frame. The task information includes the current phase of the jump $\phi$ (Sec. 4.1) and the location of the target landing position in egocentric frame. The action space includes the desired stepping frequency $f$, the desired base velocity in sagittal plane $v_x, v_z, v_\theta$, as well as the desired swing foot residuals, which are specified to different modules in the low-level controller.

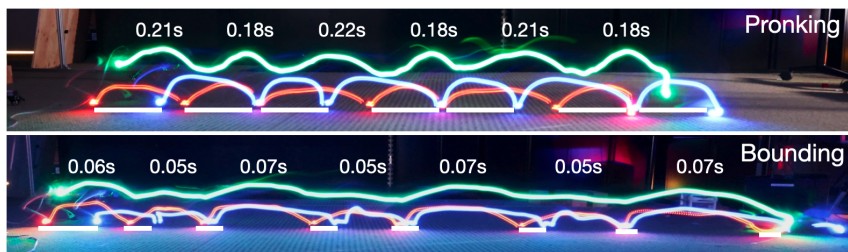

Figure 3: Long-exposure photos visualizing base (green), front foot (blue) and rear foot (red) trajectories of the robot when jumping with alternating distance commands. White lines show the foot positions during each landing (contact phase for pronking, mid-air phase for bounding). Time shows the duration of "air phase" (Fig. 2) in each jump when all legs are in the air.

**Reward Function**    We design a reward function with 9 terms. At a high level, the reward function ensures that the robot maintains an upright pose, follows the desired contact schedule, and lands close to goal. See Appendix. B.1 for the detailed weights and definitions.

**Early Termination**    To speed up training and avoid unnecessary exploration in sub-optimal states, we terminate an episode early if the robot's base height is less than 15cm, or the base orientation deviates significantly from the upright pose.

**Policy Representation and Training**    We represent policy and value functions using separate neural networks. Each network includes 3 hidden layers of $[512, 256, 128]$ units respectively with ELU activations [44]. We train our policy using Proximal Policy Optimization (PPO) [45]. Please see Appendix. B.2 for the detailed configuration.

## 6    Results and Analysis

We design experiments to validate that CAJun can learn continuous and adaptive jumping controllers. In particular, we aim to answer the following questions:

1. Can CAJun enable the robot to learn continuous jumping with adaptive jumping distances?
2. What is the widest gap that the robot can jump over using CAJun?
3. How robust is the learned jumping controller against external perturbations?
4. What is the advantage of the hierarchical design of CAJun, and what are important design choices?

### 6.1    Experiment Setup

We use the Go1 quadrupedal robot from Unitree [24], and build the simulation in IsaacGym [16, 46]. To match the GPU-accelerated simulation environment, we implement the entire control stack, including the centroidal policy and the leg controller, in a vectorized form in PyTorch [47]. We adopt the PPO implementation from `rsl_rl` [16]. We train CAJun on a standard desktop with an Nvidia RTX 2080Ti GPU, which takes less than 20 minutes to complete.

### 6.2    Continuous and Adaptive Jumping

To verify that CAJun can learn continuous, dynamic jumping with adaptive jumping distances on the real robot, we deploy the trained *pronking* and *bounding* controllers to the real robot. For each gait, we run it continuously for at least 6 jumps, where the desired jumping distance alternates between 0.3 and 1 meter. We put LEDs on the base and feet of the robot and capture the robot's trajectory using long-exposure photography (Fig. 3).

We find that both the pronking and the bounding controller can be deployed successfully to the real robot, and achieve continuous jumping with long jumping distances. Both the base and the foot trajectories exhibit clear periodicity, which demonstrates the long-term stability of the jumping controller. Moreover, the policy responds to jumping distance commands well, and results in

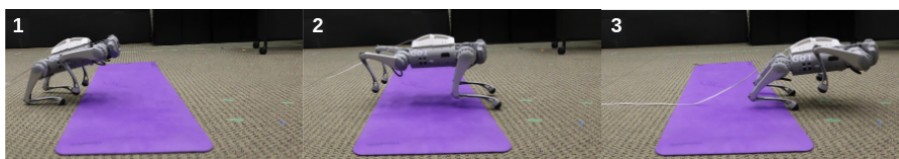

Figure 4: Using the bounding gait, the robot can jump over a 60cm-wide yoga mat without making foot contact.

| Method | Jumping Style | Widest Gap Crossed |
|---|---|---|
| TWiRL [19] | Single | 0.2m |
| Barkour [3] | Single | 0.5m |
| Margolis et al. [30] | Continuous | 0.26m |
| Walk-These-Ways [17] | Single w/ Acceleration | 0.6m |
| CAJun (ours) | Continuous w/ Adaptive Jumping Distance | **0.7m** |

Table 1: Comparison of gap-crossing capability on controllers deployed to similar-sized robots.

alternating patterns of further and closer jumps. A closer look at the duration of each air phase shows that in both the bounding and pronking gait, the centroidal policy reduces the air time by approximately 20% when switching from longer to shorter jumps. This is achieved by the stepping frequency output (Section. 4.1) of the centroidal policy. As demonstrated in previous works [22, 48], such gait adjustments can potentially save energy and extend the robot's operation time.

### 6.3 Jumping over Wide Gaps

While both the pronking and bounding gait can jump with at least 70cm of base movement in each step, we find that the bounding gait offers a unique advantage in traversing through gaps. As seen in the foot trajectories in Fig. 3, the alternating contact pattern in bounding enables the front and rear of the robot to land closely in the world frame, so that the robot can utilize *the entire jumping distance* of 70cm for gaps. To further validate this, we place a yoga mat with a width of 60cm in the course of the robot, and find that the robot can jump over it with additional buffer space before and after the jump (Fig. 4). To the best of our knowledge, CAJun is the first framework that achieves continuous jumping with such gap-crossing capability on a commercially-available quadrupedal robot (Table. 1).

### 6.4 Validation on Robustness

We design two experiments to further validate the robustness of CAJun. In the first experiment, we add a leash to the back of the robot and actively pulled the leash during jumping (Fig. 5). While both the pronking and bounding gait experienced a significant drop in forward velocity during the pull, they recovered from the pull and regained momentum for subsequent jumps. In the second experiment, we test the robot outdoors, where the robot needs to jump from asphalt to grass (Fig. 6). The uneven and slippery surface of the grass perturbed the robot and broke the periodic pattern in pitch angles. However, both policies recovered from the initial perturbation, and resume stable, periodic jumps after around 2 jumping cycles. The robustness of CAJun can be likely attributed to the high control frequency of the low-level leg controller, which enables the robot to react swiftly to unexpected perturbations, and the online adjustment of the learned centroidal policy.

### 6.5 Comparison with End-to-End RL

To demonstrate the effectiveness of CAJun's hierarchical setup, we compare it to an end-to-end RL baseline, where the policy directly outputs motor position commands. Please refer to Appendix. B.3 for the setup details. In both simulation and the real world, we run each policy for 6 jumps with a desired distance of 1 meter per jump, and report the total CoM displacement in Table. 2. While CAJun and end-to-end RL achieves comparable performance in simulation, CAJun faces a significantly smaller sim-to-real gap and outperforms e2e baseline for both gaits in the real world (25% further in bounding, 185% in pronking). We further conduct sim-to-sim transfer experiment and validate the robustness of CAJun under shifted dynamics (Appendix. B.3). While additional efforts such as domain randomization [15], system identification [31] or teacher-student training [20] could improve the robustness and reduce the sim-to-real gap for E2E methods, the hierarchical framework of CAJun offers a simple and efficient alternative that can be deployed zero-shot to the real world.

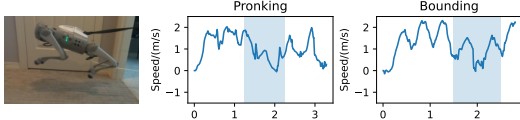
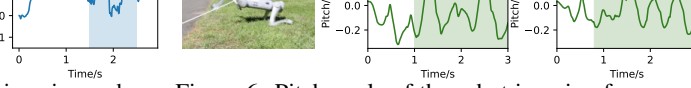

Figure 5: Forward velocity of the robot jumping under leash pulling (shaded area shows active pulling).

Figure 6: Pitch angle of the robot jumping from asphalt to grass (shaded area indicates grass)

|  | Pronking | | Bounding | |
|---|---|---|---|---|
|  | Sim | Real | Sim | Real |
| E2E | 4.17±0.01 | 1.67±0.18 | **4.61±0.03** | 3.47±0.15 |
| CAJun (ours) | **4.98±0.02** | **4.76±0.11** | 4.27±0.05 | **4.34± 0.17** |

Table 2: Total distance after 6 jumps achieved by end-to-end RL and CAJun.

## 6.6 Ablation Study

We design a set of ablation studies to validate the design choices of CAJun. We summarize the results here. Please refer to Appendix. B.4 for details.

**No Gait Modulation** The stepping frequency from the centroidal policy is essential for the stability of the robot. In *no-gait*, we disable the stepping frequency output and adopt a fixed stepping frequency of 1.66Hz for both the pronking and bounding gait, which is the average stepping frequency output from CAJun. While the baseline can achieve a similar reward, the learning process is noisy with frequent failures. Since the heuristically-designed gait might not match the capability of the robot, it is important for the policy to adjust the gait timing to stabilize each jump.

**No Swing Leg Residual** The swing residuals play a critical role in achieving long-distance jumps. To validate that, we design a baseline, *no-swing*, where we disable the swing residuals so that swing legs completely follow the heuristically-designed trajectory from the swing controller. We find that the baseline policy cannot jump as far as CAJun, and achieves a lower reward for both gaits.

**No Swing Leg Reference** The reference swing leg trajectory improves the overall jumping performance. In *NoSwingRef*, we train a version of CAJun where the centroidal policy directly specify swing foot position without reference trajectory. While *NoSwingRef* performs similarly to CAJun for the pronking gait, it jumps significantly shorter and achieves a lower reward for the bounding gait, because the bounding gait requires more intricate coordination of swing legs.

**CAJun-QP** The clipped QP in GRF optimization significantly reduced training time without noticeable performance drops. To validate this design choice, we compare the training time and policy performance of CAJun with a variant, CAJun-QP, where we solve for GRFs using the complete QP setup, where the approximated friction cone is imposed as constraints. We adopt the QP-solver from `qpth` [49], an efficient interior-point-method-based solver that supports GPU acceleration. For both the pronking and bounding gait, we find that CAJun achieves a similar reward compared to CAJun-QP. However, because CAJun-QP needs to iteratively optimize GRF at every control step, its training time is almost 10 times longer, which is consistent with prior observations [21]. Additionally, we find that the training time speed up of CAJun can be extended to other gaits such as crawling, pacing, trotting and fly-trotting. Please refer to Appendix. B.5 for more details.

## 7 Limitations and Future Work

In this work, we present CAJun, a hierarchical learning framework for legged robots that consists of a high-level centroidal policy and a low-level leg controller. CAJun can be trained efficiently using GPU-accelerated simulation and can achieve continuous jumps with adaptive jumping distances of up to 70cm. One limitation of CAJun is that, while it can adapt to changes in jumping distances, it can not land accurately at the desired location yet. This inaccuracy might be due to a number of factors such as unmodeled dynamics and state estimation drifts. Another limitation of CAJun is that it does not make use of perception, and only adjusts its jumping distances based on ad-hoc user commands. In future work, we plan to extend CAJun to incorporate perception and achieve more accurate jumps, so that the robot can demonstrate extended agility and autonomy in challenging terrains.

**Acknowledgments**

We thank He Li for helping with the motor characteristic modeling, and Philipp Wu for the design of the robot protective shell. In addition, we would like to thank Nolan Wagener, Rosario Scalise, and other friends and colleagues at the University of Washington for their support and advice through various aspects of this project.

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

# A   Details of Low-level Controller

## A.1   Notation

We represent the base pose of the robot in the world frame as $q = [p, \Theta] \in \mathbb{R}^6$. $p \in \mathbb{R}^3$ is the Cartesian coordinate of the base position. $\Theta = [\phi, \theta, \psi]$ is the robot's base orientation represented as Z-Y-X Euler angles, where $\psi$ is the yaw, $\theta$ is the pitch and $\phi$ is the roll. We represent the base velocity of the robot as $\dot{q} = [v, \omega]$, where $v$ and $\omega$ are the linear and angular velocity of the base. We define the control input as $f = [f_1, f_2, f_3, f_4] \in \mathbb{R}^{12}$, where $f_i$ denotes the ground reaction force generated by leg $i$. $r_{\text{foot}} = (r_1, r_2, r_3, r_4) \in \mathbb{R}^{12}$ represents the four foot positions relative to the robot base. $\mathbf{I}_n$ denotes the $n \times n$ identity matrix. $[\cdot]_\times$ converts a 3d vector into a skew-symmetric matrix, so that for $a, b \in \mathbb{R}^3$, $a \times b = [a]_\times b$.

## A.2   Details of the Stance Leg Controller

**CoM PD Controller**   Given the desired CoM velocity in the sagittal plane $\left[v_x^{\text{ref}}, v_z^{\text{ref}}, \omega_y^{\text{ref}}\right]$, we first find the reference pose $q^{\text{ref}}$ and velocity $\dot{q}^{\text{ref}}$ of the robot base. We set $q^{\text{ref}} = [p_x, p_y, p_z, 0, \theta, \psi]$ to be the current pose of the robot with the roll angle set to 0, and $\dot{q}^{\text{ref}} = \left[v_x^{\text{ref}}, 0, v_z^{\text{ref}}, 0, \omega_y^{\text{ref}}, 0\right]$ to follow the policy command in the sagittal plane and keep the remaining dimensions to 0. We then find the CoM acceleration using a PD controller:

$$\ddot{q}^{\text{ref}} = k_p(q^{\text{ref}} - q) + k_d(\dot{q}^{\text{ref}} - \dot{q}) \tag{9}$$

where we set $k_p = [0, 0, 0, 50, 0, 0]$ to only track the reference roll angle, and $k_d = [10, 10, 10, 10, 10, 10]$ to track reference velocity in all dimensions.

**Centroidal Dynamics Model**   Our centroidal dynamics model is based on [8] with a few modifications. We assume massless legs, and simplify the robot base to a rigid body with mass $m$ and inertia $\mathbf{I}_{\text{base}}$ (in the body frame). The rigid body dynamics in local coordinates are given by:

$$\mathbf{I}_{\text{base}}\dot{\omega} = \sum_{i=1}^{4} r_i \times f_i \tag{10}$$

$$m\ddot{p} = \sum_{i=1}^{4} f_i + g \tag{11}$$

where $g$ is the gravity vector transformed to the base frame.

With the above simplifications, we get the linear, time-varying dynamics model:

$$\underbrace{\begin{bmatrix} \dot{\omega} \\ \ddot{p} \end{bmatrix}}_{\ddot{q}} = \underbrace{\begin{bmatrix} \mathbf{I}_{\text{base}}^{-1}[r_1]_\times & \mathbf{I}_{\text{base}}^{-1}[r_2]_\times & \mathbf{I}_{\text{base}}^{-1}[r_3]_\times & \mathbf{I}_{\text{base}}^{-1}[r_4]_\times \\ \mathbf{I}_3/m & \mathbf{I}_3/m & \mathbf{I}_3/m & \mathbf{I}_3/m \end{bmatrix}}_{\mathbf{A}} \underbrace{\begin{bmatrix} f_1 \\ f_2 \\ f_3 \\ f_4 \end{bmatrix}}_{f} + \underbrace{\begin{bmatrix} 0 \\ g \end{bmatrix}}_{g} \tag{12}$$

as seen in Eq. (3).

## A.3   Reference Trajectory for Swing Legs

For swing legs, we design the reference trajectory to always keep the feet tangential to the ground, and use residuals from the centroidal policy to generate vertical movements. To find the reference trajectory, we interpolate between three key frames $(p_{\text{lift-off}}, p_{\text{air}}, p_{\text{land}})$ based on the gait timing. The lift-off position $p_{\text{lift-off}}$ is the foot location at the beginning of the swing phase. The mid-air position $p_{\text{air}}$ is the position of the robot's hip projected onto the ground plane. We use the Raibert Heuristic [43] to estimate the desired foot landing position:

$$p_{\text{land}} = p_{\text{ref}} + v_{\text{CoM}} T_{\text{stance}}/2 \tag{13}$$

| Parameter | Value |
|---|---|
| Learning rate | 0.001, adaptive |
| # env steps per update | 98,304 |
| Batch size | 24,576 |
| # epochs per update | 5 |
| Discount factor | 0.99 |
| GAE $\lambda$ | 0.95 |
| Clip range | 0.2 |

Table 3: Hyperparameters used for PPO.

where $\boldsymbol{v}_{\text{CoM}}$ is the projected robot's CoM velocity onto the $x - y$ plane, and $T_{\text{stance}}$ is the expected duration of the next stance phase, which is estimated using the stepping frequency from the centroidal policy. Raibert's heuristic ensures that the stance leg will have equal forward and backward movement in the next stance phase, and is commonly used in locomotion controllers [? 8].

Given these three key points, $\boldsymbol{p}_{\text{lift-off}}$, $\boldsymbol{p}_{\text{air}}$, and $\boldsymbol{p}_{\text{land}}$, we fit a quadratic polynomial, and computes the foot's desired position in the curve based on its progress in the current swing phase. Given the desired foot position, we then compute the desired motor position using inverse kinematics, and track it using a PD controller. We re-compute the desired foot position of the feet at every step (500Hz) based on the latest velocity estimation.

## B Experiment Details

### B.1 Reward Function

Our reward function consists of 9 terms. We provide the detail about each term and its corresponding weight below:

1. **Upright (0.02)** is the projection of a unit vector in the $z$-axis of the robot frame onto the $z$-axis of the world frame, and rewards the robot for keeping an upright pose.

2. **Base Height (0.01)** is the height of the robot's CoM in meters, and rewards the robot for jumping higher.

3. **Contact Consistency (0.008)** is the sum of 4 indicator variables: $\sum_{i=1}^{4} \mathbb{1}(c_i = \hat{c}_i)$, where $c_i$ is the actual contact state of leg $i$, and $\hat{c}_i$ is the desired contact state of leg $i$ specified by the gait generator. It rewards the robot for following the desired contact schedule.

4. **Foot Slipping (0.032)** is the sum of the world-frame velocity for contact-legs: $\sum_{i=1}^{4} \hat{c}_i \sqrt{v_{i,x}^2 + v_{i,y}^2}$, where $\hat{c}_i \in \{0, 1\}$ is the desired contact state of leg $i$, and $v_{i,x}, v_{i,y}$ is the *world-frame* velocity of leg $i$. This term rewards the robot for keeping contact legs static on the ground.

5. **Foot Clearance (0.008)** is the sum of foot height (clipped at 2cm) for non-contact legs. This term rewards the robot to keep non-contact legs high on the ground.

6. **Knee Contact (0.064)** is the sum of knee contact variables $\sum_{i=1}^{4} kc_i$, where $kc_i \in \{0, 1\}$ is the indicator variable for knee contact of the $i$th leg.

7. **Stepping Frequency (0.008)** is a constant plus the negated frequency $1.5 - \text{clip}(f, 1.5, 4)$, which encourages the robot to jump at large steps using a low stepping frequency.

8. **Distance to goal (0.016)** is the Cartesian distance from the robot's current location to the desired landing position, and encouarges the robot to jump close to the goal.

9. **Out-of-bound-action (0.01)** is the normalized amount of excess when the policy computes an action that is outside the action space. We design this term so that PPO would not excessively explore out-of-bound actions.

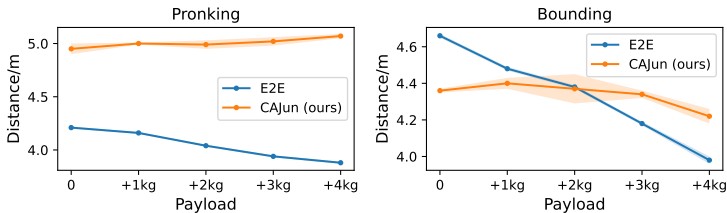

Figure 7: Comparison of total jumping distance under increased payload.

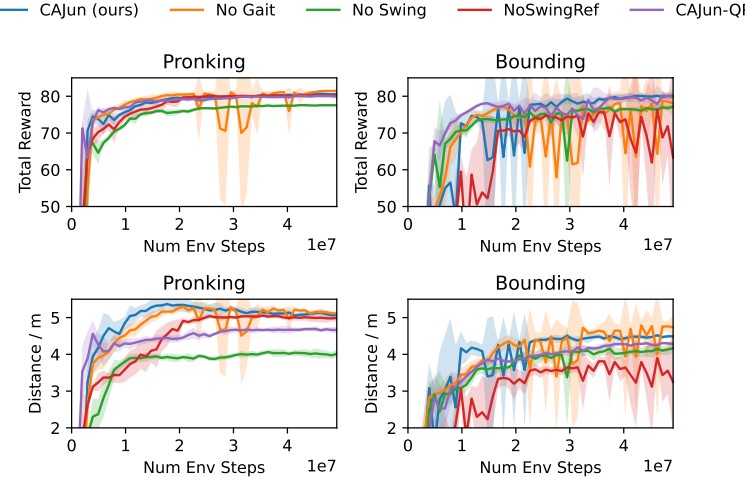

Figure 8: Reward curve and jumping distance of CAJun compared to the ablated methods.

## B.2 PPO hyperparameters

We list the hyperparameters used in our PPO algorithm in Table. 3. We use the same set of hyperparameters for all PPO training, including the CAJun policies and baseline policies.

## B.3 Comparison with End-to-End RL

**E2E Setup** We use a similar MDP setup as CAJun (section. 5) for the end-to-end RL baseline. More specifically, we use the same gait generator as CAJun to generate reference foot contacts, and include stepping frequency as part of the action space so that the policy can modify the gait schedule. However, unlike CAJun, this reference gait is only used for reward computation, and does not directly affect leg controllers. For reward, we keep the same reward terms and weights (Appendix. B.1). However, since the initial exploration phase of end-to-end RL can lead to a lot of robot failures with negative rewards, we add an additional alive bonus of 0.02 to ensure that the reward stays positive.

**Sim-to-Sim Transfer** To better understand the robustness of CAJun and end-to-end RL (E2E) under different dynamics, we conduct a sim-to-sim transfer experiment, where we test the performance of CAJun and E2E under increased body payloads. The result is summarized in Fig. 7. While the distance of E2E drops quickly with increased payload, CAJun maintains a near-constant distance even with a 4kg payload, thanks to the robustness of the low-level centroidal controller.

## B.4 Ablation Study

**Learning Curves** For each baseline, we report its total reward and CoM displacement over 6 consecutive jumps with a desired distance of 1m per jump (Fig. 8). We train each baseline using 5

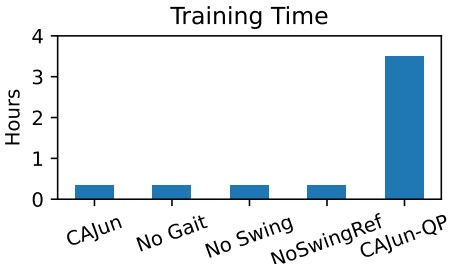

Figure 9: Training Time of CAJun compared to ablated methods.

random seeds and report the average and standard deviations. We also report the wall-clock training time in Fig. 9.

## B.5 Extension to Other Gaits

While we focus on jumping gaits in this work, CAJun is a versatile locomotion framework that is capable of learning a wide range of locomotion gaits. By adopting a different contact sequence for the gait generator (Fig.2), CAJun can learn a wide variety of other locomotion gaits such as crawling, pacing, trotting and fly trotting. With GPU-parallelization, all these gaits can be trained in less than 20 minutes. Please check our website for videos.

