# OpenReview forum: "CAJun: Continuous Adaptive Jumping using a Learned Centroidal Controller"
_robot-learning.org/CoRL/2023/Conference — CoRL 2023 Poster_

### Official Review · Reviewer_Dbjy · 2023-07-18

**Confidence:** 5
**Originality:** Fair
**Technical Quality:** Good
**Clarity Of Presentation:** Good
**Impact:** 2

**Recommendation:**

Weak Accept: I recommend accepting the paper, but will not argue for my recommendation if the majority of other reviewers have a different opinion.

**Review:**

Strengths
---------
- By presenting a working pipeline for learning jumping gaits on a widely available quadruped, the paper is interesting and relevant to parts of the audience.

- The paper is well written, and the approach is quite clear.

- The approach is sound. We can argue that the RL policy should learn to output desired endeffector/CoM trajectories, such that the least-squares solution satisfies the constraints sufficiently well to avoid detrimental effects from the clipping/scaling.

Weaknesses
----------
- There is not much novelty, because the pipeline is rather standard. The main contribution is to show that the pipeline works in this use case, not so much in the pipeline itself.

- The particular projection is not well motivated. In what sense are the projected ground forces closes to least-squares solutions?


**Quality Of The Limitations Section:**

Limitations are addressed clearly

**Questions For Rebuttal:**

- Please elaborate on how the reward is normalized (line 187). Does this normalization break Markovianity?

- Can you provide a justification for using the particular projection of Eq. 8?

- The "No Swing Leg Residual" ablation directly uses the nominal trajectory for the swing leg. I think it would be more interesting to compare to a purely learned swing leg trajectory (zero nominal). How well does the approach perform, when not providing a residual trajectory?

**Robotics Focus:**

Sufficient demonstration on hardware

**Summary Of Paper:**

The submission presents a pipeline for learning a jumping controller for a Unitree Go1 quadruped. The approach uses RL for sim2real of a higher-level controller, that outputs des. CoM velocities, step timing and residuals of the foot positions with respect to a given nominal trajectory. The des. joint torques are computed using two separate controllers: The Swing-leg controller uses IK and a PD-controller to track the desired foot trajectory; the stance-leg controller first uses a PD-controller to compute desired CoM accelerations based on the des. velocities, and then solves a QP to find des. ground reaction forces and foot Jacobian to obtain the stance leg torques. As solving the constrained QP within RL would become a bottleneck, force limits and friction cone constraints are ignored to obtain an unconstrained GP that can be solved using least-squares, and clipping/scaling the resulting forces. Two jumping gaits are learned that can jump variable distances specified as context.

**Summary Of Recommendation:**

The submission presents a pipeline for learning variable-distance jumping gaits for a lightweight quadruped and demonstrates its applicable. The novelty of the pipeline is a bit unclear, but I do think that the results are of some interest.

---

### Official Review · Reviewer_mtMy · 2023-07-19

**Confidence:** 5
**Originality:** Good
**Technical Quality:** Good
**Clarity Of Presentation:** Good
**Impact:** 3

**Recommendation:**

Weak Accept: I recommend accepting the paper, but will not argue for my recommendation if the majority of other reviewers have a different opinion.

**Review:**

Strengths:
- The reformulated stance leg control is great. I believe that the slow speed of solving this QP was a bottleneck for many works that use a hierarchical architecture [22, 23, 30, 32, 33, 35]. Accelerating it addresses an outstanding issue in this paradigm and will be useful for future works.
- Thanks to this, the 20 minutes of training time for CAJun is much faster than the prior works, both in hierarchical methods and in learned jumping [19, 3, 30].
- Comparison with prior published E2E works in Table 1 supports the work's advanced jumping performance.

Weaknesses:
- The scope of the paper's task of (continuous blind jumping) is a bit narrow. It seems like the paper's architectural improvements (swing leg residual and clipped QP training) could be extended to learning all quadrupedal gaits, which would broaden its implications.
- I noticed in Figure 3 of [17] and the corresponding subsection titled "Agile forward leap" describe crossing a gap of width 60cm using end-to-end pronking. Is that evaluation different from the others in Table 1?
- Some incomplete reporting on the ablation study (see below)

**Quality Of The Limitations Section:**

Limitations are addressed clearly

**Questions For Rebuttal:**

- The ablation study results in Appendix B.4 seem incomplete. Reporting the reward curves does not directly support the claim in section 6.6 "the baseline policy cannot jump as far as CAJun". For example, a policy with lower reward might be jumping further but with worse foot slippage or clearance, which would be important to distinguish. Similarly, the statements about the "no gait modulation" baseline in 6.6 are a bit vague -- it seems like the final reward is pretty similar. Would it be possible to evaluate each ablation by the same criteria as in Table 1 and Table 2 (continuous jumping distance and maximum jump width)?
- Can the benefit of swing leg optimization and QP clipping also be shown for trotting? I wonder if they have some benefits for standard locomotion or if they're mainly useful for jumping. Particularly for QP clipping, it could potentially strengthen the paper to emphasize that this successfully accelerates training for walking and running too.
- As mentioned above, a comparison to [17] might be missing from Table 1
- I would be interested to know the result of the following comparison. Instead of training a baseline policy with RL, you might train an end-to-end neural network with behavioral cloning to imitate the entire hierarchical controller (centroidal policy + gait generator + IK + stance controller). Such an analysis could test whether end-to-end policies are architecturally incapable of representing the same behavior as the hierarchical system which would be a pretty interesting result. Have you considered trying this / does it make sense to you?

**Robotics Focus:**

Sufficient demonstration on hardware

**Summary Of Paper:**

In this work, the authors propose learning a centroidal policy to perform blind continuous jumping with adaptive distances. The main innovations compared to prior systems are (1) allowing learning a swing leg residual for better performance and (2) reformulating the stance leg optimizer to accelerate training. The overall system is shown to jump further than prior methods.

**Summary Of Recommendation:**

This paper proposes a few nice improvements in the centroidal policy training and demonstrates a new capability (longest jump distance). The paper can be made stronger by (a) more thorough quantitative evaluation of the ablations (section 6.6), and (b) a slight expansion of scope in the demonstrated behaviors or architectural analysis.

---

### Official Review · Reviewer_JTyM · 2023-07-22

**Confidence:** 4
**Originality:** Poor
**Technical Quality:** Fair
**Clarity Of Presentation:** Fair
**Impact:** 1

**Recommendation:**

Weak Reject: I recommend rejecting the paper, but will not argue for my recommendation if the majority of other reviewers have a different opinion.

**Review:**

“Continual jumping” might be better than “continuous jumping.”

While the presented demonstrations are impressive, it is not very clear for the reviewer why the proposed technique enables wider-distance jumping compared with other end-to-end approaches. Careful analyses and discussions are expected.

As far as the reviewer knows, the technique to replace the quadratic programming under inequality constraints with a combination of unconstrained quadratic programming and postprocessing is often employed (see Hyon [*], for example), and is often criticized as it does not provide the strict optimum solution. Although the reviewer does not criticize it, at least he/she does not think it is the original contribution by the authors.

[*] Sang-Ho Hyon, Full-body compliant human-humanoid interaction: Balancing in the presence of unknown external forces, IEEE Transactions on Robotics, 23(5):884-898, 2007.


**Quality Of The Limitations Section:**

Limitations are not well addressed

**Questions For Rebuttal:**

Nothing.


**Robotics Focus:**

Sufficient demonstration on hardware

**Summary Of Paper:**

This paper addresses the quadruped locomotion control that enables continual jumping in which each hop length can be adaptively changed. It forms a hierarchical structure. The upper layer is a centroidal motion controller acquired through reinforcement learning, which outputs the swing-foot residual, the phase of the gait, and the desired velocity of the base body. In the lower layer, the swing leg controller solves the inverse kinematics, while the stance leg controller determines the desired ground reaction forces. Although the latter is formulated in general as a quadratic programming under inequality constraints, the authors omit the constraints to compute the forces in a closed form, and then clip the forces when they violate the constraints. The simplified computation, the latter runs five times as frequent as the centroidal controller, which runs every 10 ms. The controller enables wider distances of hops than an end-to-end reinforcement-learning-based controller.

**Summary Of Recommendation:**

The paper would be substantially improved if the original contribution and the lesson-to-learn of this work are clarified.

---

> ### Author Response · Authors · 2023-08-15
> **Follow up on Rebuttal**
>
> Dear Reviewer JTyM:
>
> As we are approaching the end of the rebuttal period, we are curious to know if you have any feedback on our rebuttal and updated draft. Please let us know if we have addressed your concerns and if you would like to re-evaluate our draft. Thank you.

---

### Comment · Area_Chair_nT6a · 2023-08-10
**Response to reviewers**

Dear Authors,
please submit your responses to reviewers on Openreview soon to enable a constructive discussion with reviewers during the rebuttal time window.
Best,
AC

---

> ### Author Response · Authors · 2023-08-11
> **General Response to Reviewers and Area Chair**
>
> Dear reviewers and area chair:
>
> Thank you all for your time in providing valuable feedback to our paper. We’d like to address two general questions raised by the reviewers:
>
> 1. *Comparison of CAJun with end-to-end RL (Reviewer JTyM, mTMy)*
>
> We find the smaller sim-to-real gap to be a major benefit of CAJun compared with the end-to-end RL (E2E) baseline. We train E2E using a standard PPO implementation [16] and a commonly used network architecture [16, 17, 28]. Although E2E can achieve a comparable performance with CAJun in simulation, it is less robust against shifts in environment dynamics and suffers from a larger sim-to-real gap (Table.2). In the revision, we conduct another sim-to-sim transfer experiment to further validate the robustness of CAJun compared to E2E.
>
> 2. *Concerns about novelty (Reviewer JTyM, mTMy, Dbjy)*
>
> While prior works [22, 23, 30, 32, 33, 35] have adopted a similar hierarchical architecture for locomotion control, we want to highlight two major contributions of CAJun.
> * CAJun is the first hierarchical framework that effectively utilizes GPU-accelerated simulation to reduce the training time by an order of magnitude (Fig.9). While we focus on jumping gaits in this work, this speedup of CAJun can be applied to other locomotion gaits such as walking, trotting and pacing. As shown in the updated [website](https://sites.google.com/view/continuous-adaptive-jumping/home),
> * CAJun is the first controller that achieves continuous jumping over 70cm gaps on similar-sized robots. Continuous, long-distance jumping is a challenging task for learning-based, control-based and hierarchical methods in general, and existing methods suffer from issues such as long computation time and large sim-to-real gap.
>
>
> Based on the feedback, we have posted a revision of the draft. Please find the revised paper and LaTeX diff file in the attachments of each rebuttal. Here we highlight the main changes:
>
> 1. *Sim-to-sim transfer for E2E vs CAJun*
>
> Based on the suggestions from reviewer JTyM, we conducted additional experiments of sim-to-sim transfer to compare the performance of CAJun and E2E under dynamic shifts. Please refer to Appendix B3 for details.
>
> 2. *Report of jumping distance in ablation study*
>
> Based on the suggestions from reviewer mTMy, we report both the total reward and distance jumped in the ablation study. Please refer to Fig.8 in the appendix for the updated results.
>
> 3. *Additional ablation study with no swing reference trajectory*
>
> Based on the suggestions from reviewer Dbjy, we included an additional ablation study, NoSwingRef, where the centroidal policy outputs the swing foot positions directly without reference trajectory. Please refer to Section 6.6 and Appendix B.4 for details.
>
> 4. *Result with additional gaits*
>
> In addition to the bounding and pronking gaits, we train CAJun to implement other locomotion gaits such as walking, trotting and pacing. The result can be found in Appendix B.5 and the updated [website](https://sites.google.com/view/continuous-adaptive-jumping/home).
>
> 5. *Additional Edits*
>
> Lastly, we have included additional references on constraint relaxation [39, 40, 41] based on reviewer TJyM, and included a more clear motivation for the GRF projection method based on reviewer Dbjy.
>
> Please let us know if you have any additional questions. Thank you for your time!
>
> Authors

---

### Author Response · Authors · 2023-08-11
**Rebuttal and Revision Posted**

Dear reviewers and area chair:

Sorry for the delay. We have posted a rebuttal for each review, and attached a paper revision to the rebuttals. We look forward to further discussion with the reviewers!

Authors

---

### Decision · Program_Chairs · 2023-08-30

**Decision:**

Accept (Poster)

**Comment:**

The paper proposes a learning policy method to perform blind continuous jumping with adaptive distances. The method has novelty in allowing learning a swing leg residual and accelerated training. For the camera-ready please include the changes as per reviewers' suggestions (such as additional experiments and Additional experiments).